# Emerging Ocular Side Effects of Immune Checkpoint Inhibitors: A Comprehensive Review

**DOI:** 10.3390/biomedicines12112547

**Published:** 2024-11-07

**Authors:** Kevin Y. Wu, Yoel Yakobi, Diana D. Gueorguieva, Éric Mazerolle

**Affiliations:** 1Department of Surgery, Division of Ophthalmology, University of Sherbrooke, Sherbrooke, QC J1G 2E8, Canada; 2Faculty of Medicine and Health Sciences, McGill University, Montreal, QC H3A 0G4, Canada; 3Faculty of Medicine, University of Montreal, Montreal, QC H3C 3J7, Canada

**Keywords:** oncology, immune checkpoint inhibitors (ICIs), ocular toxicity, cancer immunotherapy, immunotherapy-induced uveitis, ocular immune-related adverse events (OirAEs), tumor response, immunologic pathway, systemic immune-related side effects, combination therapy in cancer treatment

## Abstract

Immune checkpoint inhibitors (ICIs) have revolutionized cancer treatment, offering significant improvements in patient survival across various malignancies. However, their use is associated with a broad spectrum of immune-related adverse events (irAEs), including those affecting the eye and its surrounding structures, collectively termed ocular irAEs (OirAEs). Although rare, OirAEs (e.g., keratitis, uveitis, retinal vasculitis, etc.) can significantly impact a patient’s quality of life, leading to ocular complications if left untreated. This review provides a comprehensive overview of OirAEs associated with ICIs, including their clinical manifestations, underlying mechanisms, and current management strategies. We delve into the anterior and posterior segment adverse events, highlighting conditions such as dry eye, uveitis, and retinal disorders, as well as neuro-ophthalmic and orbital complications. Furthermore, we discuss the challenges in diagnosing and treating these conditions, particularly given the overlap with other autoimmune and paraneoplastic syndromes. Finally, we identify key knowledge gaps and suggest future research directions aimed at optimizing the management of OirAEs while maintaining the efficacy of cancer therapy. This review underscores the need for increased awareness among clinicians to prevent irreversible ocular damage and enhance patient outcomes.

## 1. Introduction

Immune checkpoint inhibitors (ICIs) have emerged as a cornerstone in modern oncology, transforming the treatment landscape for various cancers by harnessing the body’s immune system to target and destroy tumor cells. Despite their remarkable success, the use of ICIs is not without risk, as they can trigger a range of immune-related adverse events (irAEs) across multiple organ systems. Among these, ocular irAEs (OirAEs) represent a unique and often overlooked subset, capable of causing significant morbidity, including vision loss and other serious complications. This review aims to provide a comprehensive examination of OirAEs associated with ICIs, exploring their clinical presentations, underlying mechanisms, and current management strategies. We review adverse events affecting the anterior and posterior segments of the eye, neuro-ophthalmic conditions, and orbital complications, offering insights into their pathophysiology and the challenges they present in clinical practice. By addressing the gaps in our current understanding and highlighting areas for future research, this review underscores the importance of heightened awareness and proactive management of OirAEs to preserve visual function and optimize patient outcomes in cancer therapy.

## 2. Background on Immune Checkpoint Inhibitors

### 2.1. Mechanism of Action and Use in Cancer Treatment

Immune checkpoint inhibitors (ICIs) are a class of immunotherapeutic medications used in the treatment of various cancers. Ipilimumab was the first ICI approved by the FDA in 2011 for the treatment of metastatic melanoma, with Health Canada soon following suit and approving ipilimumab in 2012 [1,2]. Since then, a growing number of ICIs have been approved for a larger variety of solid and hematologic malignancies, including non-small-cell lung cancer (NSCLC), renal cell carcinoma (RCC), colorectal cancer, various squamous cell carcinomas, Hodgkin lymphoma, and pleural mesothelioma, among others (Table 1) [2,3,4,5,6,7,8,9].

ICIs act by inhibiting a tumor’s ability to take advantage of immune checkpoints and evade the host immune system. Immune checkpoints are negative regulators of immunity which, under normal circumstances, mediate self-tolerance and protect host tissues from autoimmune attacks. Tumor cells exploit these mechanisms by activating immune checkpoint proteins on T-lymphocytes, thereby downregulating and escaping the host immune response [1,10]. The two best-understood immune checkpoint pathways are the cytotoxic T-lymphocyte-associated protein 4 (CTLA-4) pathway and the programmed cell death protein 1 (PD-1) pathway. CTLA-4 is a transmembrane protein expressed on CD4+ and CD8+ T lymphocytes, which, when bound to the ligands CD80 and CD86 on the surface of antigen-presenting cells, leads to inhibition of cytotoxic T-cell activity and augmentation of the immunosuppressive activity of regulatory T cells. Similarly, PD-1 is an inhibitory transmembrane protein expressed on T lymphocytes, B lymphocytes, natural killer cells, and myelosuppressive dendritic cells. Its ligand, programmed cell death ligand 1 (PD-L1), is expressed on epithelial cells, dendritic cells, macrophages, and fibroblasts, and when the two bind there is an inhibition of effector T-cell proliferation and differentiation. Tumors take advantage of both pathways by expressing the ligands CD80 and PD-L1 on their cell surface. This inhibits T lymphocytes in the tumor microenvironment from mounting an immune response and prolongs their own survival and proliferation [1,10]. ICIs inhibit these interactions and, in doing so, promote T-lymphocyte activation to induce an immune-mediated attack against the tumor (Figure 1).

PD-1 and CTLA-4 are immune checkpoints expressed by T cells that inhibit immune responses. Tumor cells can escape immunity by interaction with their ligands and avoid immune recognition. Immune checkpoint inhibitors (such as anti-CTLA-4, anti-PD1 and anti-PD-L1) are monoclonal antibodies which work by activating immune responses, thus halting tumor evasion. The resulting increased immune response can lead to a variety of immune-related adverse events. 

There are three classes of ICIs in widespread use, each of which augments the anti-tumor response by blocking either the receptor or the ligand in one of the aforementioned pathways: anti-CTLA-4 ICIs, such as ipilimumab and tremelimumab; anti-PD-1 ICIs, such as nivolumab, pembrolizumab, and cemiplimab; and anti-PD-L1 ICIs, such as atezolizumab, avelumab, and durvalumab (Figure 2) [2,3,4,5,6,7,8,9]. All three classes have demonstrated significant efficacy in cancer treatment, improving patient outcomes in terms of Progression Free Survival and Overall Survival [11,12,13,14]. Treatment regimens include both monotherapy and combination therapy, in which ICIs targeting the CTLA-4 pathway and the PD-1 pathway target two axes of the immune checkpoint blockade simultaneously. Such therapy results in better outcomes compared to monotherapy alone; in one study comparing combination therapy to monotherapy, response rates and survival increased by 58% and 11.5 months, respectively [1,12,15].

### 2.2. Side Effects of Immune Checkpoint Inhibitors

While a blockade of immune inhibition is beneficial in treating cancer, it can also lead to off-target immune-mediated attacks in almost every organ and tissue, known as immune-related adverse events (irAEs). Compared to more conventional cancer treatments like chemotherapy and radiotherapy, side effects from ICI use are more varied. The most common irAEs manifest in the dermatologic (rash, pruritus), gastrointestinal (diarrhea, colitis), and endocrine (thyroid dysfunction, pituitary inflammation, adrenal insufficiency) systems, but ICI toxicity is also known to present more rarely as pneumonia, myocarditis, neurotoxicity, myositis, nephritis, and in other organ systems including the eye and its surrounding structures (Figure 3 and Figure 4). Additionally, just as combination therapy is associated with better tumor response rates, so too is it associated with an increased number and severity of irAEs [10,16].

While ocular irAEs (OirAEs) are rarely life-threatening or result in death, they can cause significant deteriorations in patient quality of life. Initially believed to occur in less than 1% of patients on ICI therapy, the current literature suggests that their incidence ranges between 2.8 and 4.3%. However, OirAEs may be generally underestimated due to insufficient documentation—a result of research studies being limited to case reports and case series, until recently [17,18,19]. Regardless of its true value, such low incidence predisposes OirAEs to being overlooked by patients and clinicians, increasing the risk of irreversible damage and vision loss by the time patients present to the ophthalmologist. For example, mild uveitis can result in increased intraocular pressure, synechiae, and macular edema; retinopathy can lead to intraretinal fluid entrapment and photoreceptor loss; orbital inflammation can cause chronic pain and optic nerve compression; and ocular surface disease can lead to spontaneous corneal perforation [20,21,22,23]. Thus, patients are at further risk of clinical deterioration and reduced medication compliance, perpetuating further reductions in cancer survival [1,24,25]. Such a vicious cycle of worsening outcomes highlights the need for routine screening of OirAEs, which is severely lacking. With the expansion of ICI use, more comprehensive reporting of OirAEs is to be expected, leading to more thorough understandings of ophthalmic side effects for the practicing ophthalmologist and oncologist [26].

The National Cancer Institute has classified irAEs into five grades of increasing toxicity—Grade 1, mild or asymptomatic; Grade 2, moderate; Grade 3, severe; Grade 4, life-threatening; and Grade 5, death [10]. Mazharuddin et al. have adapted this grading system to address ocular irAEs specifically (Table 2). While there are no clear clinical practice guidelines tailored to OirAE management, Shahzad et al. have suggested a progressive management strategy based on common trends in the literature (Figure 5). Overall, management largely consists of deciding whether to discontinue ICIs, whether treatment with corticosteroids is indicated, and whether adjunctive treatments such as immunomodulators and immunosuppressants should be considered [17,26].

### 2.3. Potential Mechanisms of irAEs in Ocular Tissues

The underlying mechanisms of OirAEs are as of yet incompletely understood. Under normal conditions, the eye benefits from an immune privilege, meaning that the intraorbital environment has natural protection from inflammation-mediated damage. This privilege results from the blood–retina barrier, the absence of lymphatic vessels, and innate inhibitory costimulatory molecules [27]. These molecules, which include PD-L1, are expressed on the cornea, stromal cells, iris–ciliary body, and retinal pigment epithelium (RPE). They facilitate controlled immunosuppression of the ocular environment, preventing both inflammation and autoreactivity in the eye. While CTLA-4 is not directly involved in the immune privilege of the eye, it is responsible for more upstream regulation of cytotoxic T-cell activity. It follows that blockade of these molecules and pathways could lead to ophthalmic adverse events [27,28,29,30,31,32].

There are two accepted mechanisms by which ICI therapy triggers immune-mediated attacks, both of which result from a disruption between self-tolerance and immunity (1) via the induction of direct immune cell-mediated attacks and (2) via exacerbations of paraneoplastic syndromes [24,33,34]. Regarding (1), these represent irAEs that could arise as a result of general disinhibition of the immune system, even in the absence of a neoplasm. Adverse events in this category are generally believed to be caused by direct off-target attacks by T cells on healthy cells [35,36]. For example, in vitro studies of corneal tissue have demonstrated that PD-L1 prevents T-cell-mediated destruction of corneal endothelial cells [37]. Regarding (2), these represent inflammatory syndromes that can arise in cancer patients and which can be exacerbated when ICI therapy is initiated. Paraneoplastic syndromes are not caused by the tumors themselves but rather are believed to arise from cross-reaction of antibodies between tumor antigens and self-antigens. With the administration of ICIs, there is a suspected increase in the production of these cross-reactive antibodies and thus an augmented inflammatory response [38,39]. Both aforementioned mechanisms are implicated in the manifestation of OirAEs.

Of note, ICI toxicity can often mimic auto-inflammatory and auto-immune diseases [40]. It is unclear whether these presentations are exacerbations of pre-existing diseases or if they result from distinct ICI-induced autoimmune-like manifestations, but it is likely that both pathways contribute to the development of irAEs. Further, the literature has seemingly accepted that tumors themselves can play a role in inducing autoimmunity via tumor-associated immune disturbances and the release of self-antigens, but their precise involvement in the mechanism of OirAEs remains unclear [41].

In this comprehensive review, we will discuss the clinical presentations, potential mechanisms, customary management, and gaps in literature relating to common OirAEs following anti-CTLA-4, anti-PD-1, and anti-PD-L1 therapy. Of note, cancer treatment has grown to include other ICI regimens, among them the anti-lymphocyte activation gene-3 (LAG-3) ICI known as relatlimib, but there are as of yet no reports of OirAEs associated with these treatments and they will thus not be discussed in this review [10].

## 3. Anterior Segment Adverse Events

The most common reported adverse events associated with the anterior segment include dry eye and uveitis.

### 3.1. Corneal Disorders

Dry eye is the most common OirAE with a reported incidence between 3% and 24% [42]. It presents as ocular inflammation and irritation due to reduced tear quality, tear-film instability, and tear hyperosmolarity [19]. In an American retrospective study evaluating 996 patients on ICI therapy for ocular side effects, 16 of the 28 patients who manifested OirAEs suffered from some form of dry eye [25].

Several immunopathologic mechanisms have been proposed to explain the relationship between ICI therapy and dry eye [43]. One theory suggests that blockade of the PD-1 pathway disrupts the immune homeostasis of the eye and leads to the infiltration of T cells into the cornea [44]. A second hypothesis posits that dry eye is the result of suppressed self-tolerance from ICIs resulting in lacrimal gland dysfunction and sicca syndrome [45]. A third mechanism seems to involve the infiltration of CD8+ T cells on the corneal surface and the secretion of IL-2 cytokines, leading to sarcoidosis-like granulomatous infiltration of the lacrimal glands [46]. The end result of these mechanisms is a disruption in the delicate checkpoint processes of the eye, an imbalance in the innate and adaptive immune cells, and the manifestation of dry eye disease [47]. Studies analyzing the mechanisms of innate and adaptive immunity implicated in the pathogenesis of ICI-induced dry eye disease are still expanding [48].

Management of ICI-induced dry eye disease involves the application of artificial tears (preferably lipid-based for enhanced tear film stability) and topical cyclosporine (0.05% Restasis and 0.09% Cequa, both commercially available), a downregulator of T-cell activity [49]. In addition to these treatments, other usual conservative and pharmacological therapies for dry eye management, similar to those used in typical dry eye cases, may also be considered. Of note, documented cases report worse prognosis in patients non-compliant to cyclosporine therapy, including instances of corneal perforation [44].

Corneal graft rejection has also been described with ICI therapy, which is related to the aforementioned disruption in the immunosuppressive environment of the cornea but is also suggested to involve a reduced capacity to trigger T-cell apoptosis. In cancer-free mouse models that underwent corneal transplantation, blockade of the PD-1 pathway led to corneal graft rejection, which supports this hypothesis [37,50]. Most reports involving corneal graft rejection responded to systemic corticosteroids.

Overall, current management of ICI-associated corneal side effects stems from expert opinion, case studies, and retrospective studies. Prospective clinical trials are still needed to provide concrete insight into management.

### 3.2. Uveitis and Its Various Subtypes

Uveitis is another commonly reported OirAE, occurring in about 1% of patients at a median of 9 weeks from ICI initiation [51]. Anterior uveitis is the most common subtype of uveitis among patients on ICIs, followed by panuveitis and posterior uveitis [52,53]. Clinical manifestations include blurry vision, conjunctival erythema, photophobia, and ocular pain.

Other ICI-induced uveitis include birdshot-like uveitis, undifferentiated uveitis, choroiditis, and bilateral diffuse melanocytic proliferation [54]. A specific subtype of ICI-induced uveitis known as Vogt-Koyanagi-Harada (VKH)-like uveitis, characterized by exudative retinal detachments, has been reported with patients on anti-PD-1 therapy [55,56]. The majority of cases involve patients with malignant melanoma, but it has also been reported in association with ovarian cancer [57,58,59,60,61,62,63].

It is thought that VKH-like uveitis results from cross-reaction between malignant melanoma antigens and antigens on self-melanocytes, resulting in T-cell-mediated attacks against the host choroidal body [51]. This pathophysiology is similar to that of sympathetic ophthalmia, where immune responses target ocular melanocytes following trauma. Most cases were successfully treated with intravenous corticosteroids and steroid pulse therapy [64,65]. It is unclear at present if ICIs should be discontinued in VKH-like uveitis, relegating this decision to individual clinical judgment [36].

Regarding the other subtypes of ICI-induced uveitis, the precise pathophysiological mechanisms have yet to be elucidated, but several trends have been noted in both ICI therapy and cancer type. The overall risk for uveitis significantly increases after 1 year [66], and a strong association is linked to nivolumab, a PD-1 inhibitor [23,67,68], which accounted for 32% of total uveitis cases but is most strongly associated with ipilimumab, a CTLA-4 inhibitor [23]. There is also high incidence when the two treatment arms are combined [53]. Recent retrospective pharmacovigilance studies found similar conclusions [18,69].

Further, ICI-induced uveitis seems to occur disproportionately more in melanoma compared to other cancer types [18,70,71]. This suggests that uveitis risk may be influenced by differences in tumor type and microenvironment [18,72]. Another proposed mechanism is the possible development of increased immunogenicity in melanoma patients and may also be related to the aforementioned cross-reactivity between antibodies targeting melanoma antigens and self-antigens [21,73].

Based on the American Society of Clinical Oncology guidelines, the treatment of ICI-induced uveitis depends on the severity of presentation [74]. Uveitis linked to CTLA-4 inhibitors is usually of mild grade, as is anterior uveitis. In these cases, temporary withdrawal of ICI therapy and the administration of topical corticosteroids, cycloplegic agents, or systemic corticosteroids are recommended, but most of these cases resolve entirely with corticosteroids without ICI withdrawal [36,75,76,77]. On the other hand, panuveitis and posterior uveitis tend to necessitate more aggressive therapy with a course of topical, periocular, intravitreal and systemic corticosteroids [78]. Permanent discontinuation of ICI therapy is recommended in severe cases [79,80]

## 4. Posterior Segment Adverse Events

Retinal disorders are among the rarer OirAEs. In their pharmacovigilance analysis of the FDA Adverse Event Reporting System (FAERS) database, Bomze et. al. 2022 identified 41,674 patients treated with ICIs, and, of the 3% who developed OirAEs, only 10.7% of those presented with retinal disorders [18]. Consequently, there is a substantial gap in research into retinal irAEs, and the studies that do exist are limited in their scope, power, and generalizability, either due to their small sample sizes or their reliance on case series and case reports [20,23,81]. As it stands, retinal disorders seemingly occur more often in cases of melanoma, likely due to the retina’s close association with the choroid, which we have established is subject to immune-mediated attacks given the cross-reactivity between choroidal self-melanocytes and malignant melanoma cells [21]. Further research is warranted to confirm this theory and better characterize the incidence of retinal irAEs across cancer types. Details on retinal OirAEs can be found in Table 3.

### 4.1. Retinopathies and Maculopathies

The best understood retinal disorders are the autoimmune retinopathies and maculopathies, largely due to their status as paraneoplastic disorders and consequent capacity to arise even in the absence of ICI therapy. These disorders consist of melanoma-associated retinopathy (MAR), cancer-associated retinopathy (CAR), and acute exudative paraneoplastic vitelliform maculopathy (AEPVM), each of which results from molecular mimicry and cross-reaction between neoplastic antigens and various retinal proteins. They present with an overlapping symptomatology, namely painless visual loss or blurriness, dysphotopsia, nyctalopia, metamorphopsia, and micropsia, with an onset from anywhere from two weeks of initial ICI administration to within two years [84]. Of note, symptoms are much subtler in AEPVM compared to MAR and CAR. These disorders can be differentiated by their associated cancers and antibody profiles; MAR presents only in cases of melanoma, CAR mostly in cases of carcinoma and SCLC and some melanoma cases, and AEPVM in both. Studies are yet to be performed to ascertain the true incidence and prevalence of these disorders in each cancer type [85].

Though there is consensus regarding the basic pathophysiology of these disorders, the interaction between the paraneoplastic effect and the contribution of ICIs has yet to be established. At its simplest, these disorders result from the typical paraneoplastic disorder-associated cross-reaction between antibodies and self-antigens; in MAR, there is an immune-mediated attack against ON bipolar cells; in CAR, against photoreceptor cells; and in AEPVM, against RPE cells (Figure 6). Notably, all three irAEs share overlapping antibody profiles despite differences in their targeted anatomical structure, warranting further research into describing what unique pathophysiological mechanisms lead to each disorder. In any case, since these antibody-mediated attacks can occur in the absence of ICI administration, it is posited that the ICI-associated disinhibition of the immune system can both induce these paraneoplastic effects or exacerbate them from a subclinical to a clinical level. However, autoantibodies have been found to be positive in only 50–65% of patients, suggesting that there are other mechanisms at play in manifesting these adverse events [83,85,101,102]. One group of researchers have suggested that ICIs can impact the integrity of the retina and choroid in the absence of any neoplasm (and thus in the absence of any paraneoplastic effect); by injecting mouse models with ipilimumab, Mukai et. al. found OCT- and ERG-based disturbances in the structure and function of the outer retina. On histology, they demonstrated staining of photoreceptor structures and invasion of CD8+ and CD45+ cells into the retina and choroid, respectively, altogether suggesting that anti-CTLA-4 ICIs induce immune-mediated attacks against the photoreceptor layer—the primary target of CAR [103]. Though these results are relatively recent, they suggest that there may be multiple pathways behind the pathophysiology of CAR, at the very least. Further research is needed to determine whether ICIs targeting the PD-1 pathway can induce similar attacks against photoreceptors, whether bipolar cells and RPE cells can also suffer lymphocyte-mediated attacks triggered by ICIs, and how these lymphocyte-mediated attacks might interact with the antibody-mediated attacks of the aforementioned paraneoplastic retinopathies and maculopathies.

There are further irAEs that might fit into this category of adverse events but have yet to be as fully characterized, such as MAR, CAR, and AEPVM. For example, acute macular neuroretinopathy (AMN) is also suspected to result from immune-mediated inflammation of pericentral photoreceptors leading to central and pericentral scotomas and reductions in visual acuity. However, descriptions of ICI-induced AMN are rare in the literature, given that it has only been associated with atezolizumab use. Autoantibody profiles for AMN have yet to be described [89].

Treatment for AEPVM is more straightforward than for MAR and CAR and likely for AMN as well. The immune-mediated damage in AEPVM is known to be reversible with good prognosis, and thus the literature suggests prioritizing tumor control and continuing ICI administration. Resolution of AEPVM would follow reduction in tumor burden. MAR, CAR, and AMN, on the other hand, confer permanent retinal damage and vision loss as a consequence of the retina’s limited capacity for regeneration. Physicians can choose to continue ICI administration in the hopes that eventual tumor control will reduce irAE progression, but such a dramatic loss of vision with poor prognosis incentivizes physicians to withdraw ICIs and opt for immunosuppressive and immunomodulatory therapy. However, immunosuppressive or immunomodulatory treatments carry the potential drawback of weakening the immune response, possibly reducing ICI efficacy against the tumor and increasing the risk of infections. No studies have yet been performed to quantify the risks and benefits of continuing ICI therapy in the context of irreversible retinopathies [16,24,36,85,104,105,106].

### 4.2. Other Retinal Disorders

Other ICI-induced retinal pathologies include retinal vasculitis, choroidal neovascularization, macular edema, retinal detachment, and fundus depigmentation. Many of these irAEs most often co-present with other OirAEs, such as one or more of the various subtypes of uveitis, but they are overall much rarer, to the extent that little is known regarding their incidence and pathophysiology. Treatment generally follows the same principles of steroid therapy with or without ICI withdrawal and adjunctive immunomodulation.

## 5. Neuro-Ophthalmic Adverse Events

Neuro-ophthalmic disorders are among the most common ocular irAEs, falling just behind dry eye and uveitis, and can affect a broad range of structures, from supranuclear pathways to cranial nerves and neuromuscular junctions (Table 4). In their abovementioned pharmacovigilance analysis, Bomze et. al. found that 30.8% of patients with OirAEs presented with some form of vision disorder, comprised of blurred vision, visual impairment, diplopia, and blindness, and 8.4% presented with an optic nerve disorder [18,107]. Comparatively, in a review of 290 published cases, Martens et. al. found that 24.5% of patients with OirAEs had some form of neurological disturbance. These patients predominantly presented with lung cancer and were treated with CTLA-4 blockade. Seeing as lung cancers are commonly treated with ipilimumab, a CTLA-4 inhibitor, it remains unclear whether cancer type or ICI regimen plays a larger role in increasing the risk of neuro-ophthalmic irAEs [36]. Further research is warranted.

Current understanding of neuro-ophthalmic irAEs is limited by the same factors limiting our understanding of other OirAEs, namely small sample sizes, reporting bias in published case reports, and a scarcity of data on patient evolution over time. However, neuro-ophthalmic irAEs introduce a further confounding factor in that many of their presentations—papilledema, optic neuritis, cranial neuropathies, and internuclear ophthalmoplegia, among others—are known signs and symptoms of central nervous system metastases. Seeing as such metastases are often unidentified or unreported in registry populations, we lack the necessary evidence to establish a causative link between ICI administration and the development of neuro-ophthalmic irAEs, rendering our estimates of incidence and our pathophysiological theories less reliable [70,107,120,123].

### 5.1. Optic Nerve Disorders

Many studies, reviews, and case reports label optic nerve pathologies as optic neuropathy, which is a broad category of disease that fails to capture a specific diagnosis. The term “optic neuropathy” merely indicates that there is damage to the optic nerve without indicating whether said damage corresponds to optic neuritis, optic nerve ischemia, or neuromyelitis optica spectrum disorder (NMOSD) [124]. Future research should strive to better characterize the form of optic neuropathy affecting patients; in the meantime, the symptom profile of ICI-induced optic neuropathy commonly reported in the literature seemingly represents optic neuritis.

ICI-induced optic neuritis is invariably distinct from idiopathic optic neuritis. Where the idiopathic form is characterized by the classic triad of unilateral vision loss, dyschromatopsia, and pain, ICI-induced optic neuritis more commonly presents as bilateral and painless vision loss with less frequent changes to colour vision. The bilateral presentation of the disease is easily explained by the systemic nature of the treatment, but the painlessness and lack of colour vision changes are less understood. In a retrospective analysis of eleven patients with ICI-induced optic neuritis, only two were found to present with pain; similar trends are apparent in other case series, but these observations are confounded by concomitant OirAEs such as anterior uveitis, which is known to present with ocular pain [125]. Pain in idiopathic optic neuritis is often attributed to demyelination of the optic nerve, which is visualized as perineural enhancement on MRI, but such enhancement tends to spare the retrobulbar and proximal segments of the nerve in the ICI-induced form and instead localizes to the intraorbital segment. Such proximity to mechanical disruption by the extraocular muscles would be expected to induce pain, further obfuscating our current understanding of this OirAE. Of note, not all patients diagnosed with ICI-induced optic neuritis undergo diagnostic MRI, and not all those that do undergo it demonstrate perineural enhancement [70,121,122,125]. While there are likely multiple mechanisms at play, researchers would benefit from quantifying how many cases of enhanced and non-enhanced MRI lesions present with pain to determine whether demyelination is a contributing factor towards pain in ICI-induced optic neuritis to begin with. If there is a clear distinction in the clinical presentation of enhancing and non-enhancing lesions, the latter might be considered as a distinct diagnosis altogether.

The pathophysiology of ICI-induced optic neuritis is also yet to be elucidated. This form of the disease tends to be seronegative and lacks the typical antibodies present in the idiopathic form, namely anti-myelin oligodendrocyte glycoprotein and anti-aquaporin 4. Though ICI-induced optic neuropathy has been found to be associated with anti-optic nerve antibodies, these findings can scarcely be generalized to other optic nerve disorders given the aforementioned diagnostic obscurity in labelling an adverse event “optic neuropathy”. Some researchers have proposed that CTLA-4 blockade could induce an upregulation of costimulatory T-cell activation pathways that result in demyelination due to similar mechanisms being documented in animal models, but these mechanisms have yet to be validated in humans [84,122,126,127,128]. This theory is also limited by the fact that not all cases of ICI-induced optic neuritis present with demyelination, as previously mentioned, meaning other processes are involved in this unique clinical presentation. Further research is warranted to better understand the pathophysiology of ICI-induced optic neuritis.

Little evidence is available to determine optimal treatment regimens for ICI-induced optic neuritis. Given the fact that this OirAE involves a mild form of visual loss and overall good prognosis, some believe that ICI withdrawal may be unnecessary, but further research is required to quantify the risks and benefits of such an approach [121,129].

### 5.2. Movement Disorders

Myasthenia gravis (MG) is unique among OirAEs in that it has a primarily ocular presentation but also carries a significant risk of systemic effects and death. Ocular symptoms—comprised mostly of ptosis and diplopia—reportedly occur in anywhere between 50% and 91% of ICI-induced MG patients, with 16.4% of patients presenting with isolated ocular MG. The remainder of patients present with typical MG signs and symptoms, such as dysarthria, dyspnea, and limb weakness, though with a larger propensity for bulbar symptoms compared to idiopathic MG [24,112,130]. The high risk of mortality in ICI-induced MG relates to a higher risk of rapidly progressive respiratory depression compared to idiopathic MG, but also to its rare subtype known as Myositis, Myocarditis, and Myasthenia Gravis Overlap Syndrome (IM3OS). Up to 30% of ICI-induced MG cases are associated with myocarditis and up to 40% with myositis. On its own, ICI-induced MG is linked to a 20-40% mortality rate, a 21% mortality rate when overlapping with myositis, and a mortality rate up to 50% when overlapping with myocarditis [107,112,115]. A major challenge that may explain such high mortality is the atypical antibody profile of ICI-induced MG.

While the pathophysiology of idiopathic MG is widely accepted to involve an autoantibody-mediated attack against anti-acetylcholine receptors, many cases of ICI-induced MG are seronegative. Since physicians rely on antibody assays and electromyography to diagnose MG, this eliminates a valuable diagnostic feature of the disease and points to the importance of developing new diagnostic and screening tools. First, identification of patients at higher risk of ICI-induced MG can be valuable, including analyzing both the demographic and therapeutic trends in presentation; for instance, MG-associated ophthalmoplegia is more prevalent in Asian patients with ICI-induced MG compared to White patients (43.75% versus 12.77%, respectively), and nivolumab has been found to be more associated with positive anti-AChR antibody assays than pembrolizumab despite both being PD-1 inhibitors [23,24,112,118,130]. After identifying patients at higher risk, screening should encompass more detailed ophthalmic exams; as stated earlier, ocular symptoms are among the most common in patients with ICI-induced MG, and thus identifying patients with ptosis and variable forms of diplopia (i.e., those that lack the characteristic signs of an affected cranial nerve pair) would go a long way in ensuring patients at risk of progression to IM3OS and clinical deterioration receive proper care [107].

Researchers have speculated that a secondary pathway of MG induction could be related to direct infiltration of T-lymphocytes into muscle fibres, but such theories have not yet been validated nor are there diagnostic tests by which to detect these mechanisms [107,131].

Unlike other OirAEs, treatment of ICI-induced MG is more reliant on immunomodulators than high-dose steroids. While there is literature describing cases successfully treated with steroids, their use has also been associated with MG exacerbations requiring additional and more invasive treatment, particularly in cases of IM3OS and MG refractory to steroids. Example alternatives include IV immunoglobulins, plasmapheresis, and even tacrolimus, infliximab, and mycophenolate mofetil. Supportive and symptomatic therapies, such as pyridostigmine and supplemental oxygenation, should also be administered. Treatment guidelines for these conditions have yet to be established [107,109,112,114,115].

### 5.3. Other Neuro-Ophthalmic Disorders

Most other neuro-ophthalmic irAEs are rare and have only been recorded in case reports and case series, including optic atrophy, NMOSD, papilledema, central retinal artery occlusion, branch retinal vein occlusion, cranial nerve palsies, Lambert–Eaton Myasthenic Syndrome, giant cell arteritis, and opsoclonus–myoclonus–ataxia syndrome. No records have been found regarding ICI-associated trochlear nerve palsies, with ocular cranial nerve palsies being limited to the oculomotor and abducens nerves. Ophthalmoplegia has also been recorded as a distinct irAE despite often being a symptom of larger diseases, such as myasthenia gravis or thyroid-like orbitopathy, which will be discussed below. Future research should aim to more accurately classify patient presentations [24,45].

## 6. Orbit and Ocular Adnexa Adverse Events

Orbital adverse events are also reported in association with ICI use. A summary of orbital OirAEs can be found in Table 5.

### 6.1. Thyroid-like Orbitopathy

Thyroid-like orbitopathy is an autoimmune inflammatory disorder affecting ocular and orbital tissues. It is rarely reported as an adverse event, with its most commonly-associated symptoms being ophthalmoplegia and proptosis [45]. While most cases are associated with the administration of CTLA-4 blockade [132,133,134,149], it has also been described in association with nivolumab [134], pembrolizumab [135,136], and with combination therapy [134,137].

The pathophysiology of thyroid-like orbitopathy is believed to be related to the role of CTLA-4 blockade in the activation and proliferation of CD8+ T lymphocytes and the resultant exacerbation in inflammation, which would explain the higher incidence of this disease with anti-CTLA-4 regimens [133]. Studies examining the orbital tissue of patients with thyroid-like orbitopathy found that lower expression of CTLA-4 was correlated with a higher burden of disease [150]. Most cases are treated with either systemic high-dose steroids with tapering to oral steroids, or with observation, leading to symptom resolution. The literature has demonstrated no indication for ICI withdrawal [134,151].

### 6.2. Inflammatory Orbitopathy

Inflammatory orbitopathy has been described mostly in association with ipilimumab and pembrolizumab therapy and tends to be treated with high-dose steroids [49,152,153,154]. However, the duration of treatment is variable and there is no consensus on guidelines or management, warranting further studies. It is important to retain inflammatory orbitopathy in the differential diagnosis of ocular inflammatory conditions associated with ICI therapy as a distinct entity from thyroid-associated orbitopathy.

### 6.3. Orbital Myositis

Cases of orbital myositis following ICI therapy have been described, which were effectively treated with high-dose oral steroids and, in more severe cases, a combination of methylprednisolone, mycophenolate mofetil, and IV immunoglobulin [16,134,138,139,141,142,143,144,145]. Recently, a case of bilateral orbital inflammation with tendon-involving myositis was described following ipilimumab rechallenge, which was successfully managed with IV corticosteroid and immunoglobulin therapy [140]. This case is noteworthy as it illustrates that drug rechallenge can be exercised with caution when adjunctive treatment measures are considered. The pathophysiology of ICI-induced orbital myositis remains unclear.

### 6.4. Orbital Apex Syndrome

Few cases of orbital apex syndrome (OAS) have been reported. One such case resembling OAS has been described in association with pembrolizumab in a patient that presented with multiple cranial neuropathies. Since the patient did not exhibit pain, treatment involved steroid therapy alone, but this case serves as a useful reminder that the treatment of ICI-induced cranial neuropathies should involve a combination of observation, prednisone, and—in cases of pain—gabapentin, pregabalin, or duloxetine. This demonstrates that symptomatic treatment, including optimizing pain control, needs to be considered in the management of OirAEs [147]. A single other case of OAS was found in the literature which involved ipilimumab and was effectively treated with steroid therapy [146].

While the pathophysiology of ICI-induced OAS remains unknown, its corresponding idiopathic forms are usually caused by systemic inflammatory diseases or generalized orbital inflammation [155]. Researchers postulate that ICI therapy may lead to inflammation of nerves and neurotoxicity, both of which would contribute to OAS-like symptoms [147]. An interaction between inflammatory and immune mechanisms may also be involved [146].

## 7. Future Directions

Understanding whether different toxicities have distinct immunologic origins is of crucial importance, as is clarifying how baseline immunologic differences play a role in the risk and development of irAEs, if at all [156]. We believe this to be particularly relevant for OirAEs, seeing as their diverse presentations can affect all segments of the eye with serious consequences if left untreated. Though there appear to be trends in the ocular toxicities associated with each ICI treatment regimen, further research is needed to clarify and quantify the risks of each ICI in order to optimize management and patient outcome [157].

Though predictive risk scores for more common irAEs exist, there are none for OirAEs, nor are there any reliable biomarkers to predict OirAEs and their severity. Potential irAE biomarkers include autoantibodies, T-cell and B-cell biomarkers, and microbiome and genomic biomarkers, but, to our knowledge, no studies have examined ocular biomarkers specifically [40,158,159,160]. We believe that continued research into autoantibodies for irAE biomarkers will be of crucial relevance to the eye and we agree with the literature that future stratification tools for irAE-related risk assessment would involve clinical, pathologic, genetic, and protein expression data [108].

The presence of irAEs themselves has been proposed as a predictive biomarker for treatment effectiveness, as they have been associated with better patient outcomes in some studies. It would be interesting to examine whether such a trend exists in the eye, being an immune-privileged organ [110,161].

Potential individual-level factors such as age, ethnicity, sex, and pre-existing comorbidities may impact irAE development and progression, but data are limited [41]. No studies to date have specifically examined the role of ethnicity in the development of OirAEs [24]. Studies using the FAERS database use data which stem mainly from America, Europe, and Japan, limiting the sample of ethnicities from which to draw conclusions [162]. Regarding potential sex differences, a comprehensive longitudinal study analyzing data from the FAERS database found that male patients who developed irAEs had worse prognosis and higher mortality in all ICI categories compared to their female counterparts. These researchers did not detect a signal specific to OirAEs, but it is likely that incomplete data and underreporting could have impacted the study’s findings [162]. Further studies clarifying the relationship between sex and risk of irAEs are warranted, especially in the realm of OirAEs given the propensity for many ocular autoimmune conditions to predominantly affect women.

Finally, future research should aim to elucidate how to best optimize the treatment of irAEs and OirAEs without compromising cancer treatment outcomes. In light of a growing body of evidence suggesting that irAEs may be correlated with a better response to ICI therapy, prospective studies should aim to characterize the optimal management of OirAEs and their impact on patient outcomes, including in the context of ICI rechallenge studies [163].

Many knowledge gaps remain regarding the understanding of irAE mechanisms, the impact of ICI class on irAEs, potential risk factors, the role of auto-immunity in their development, and optimization of management [156]. A brief review of the remaining questions pertinent to OirAEs can be found in Table 6.

## 8. Conclusions

Immune checkpoint inhibitors (ICIs) have transformed oncology by enhancing survival rates across various cancers, but they also bring the risk of immune-related adverse events, including those affecting the eye. Ocular irAEs (OirAEs), though uncommon, can lead to significant visual impairment and decreased quality of life if not promptly recognized and managed. This review has highlighted the diverse range of OirAEs, including anterior and posterior segment disorders, neuro-ophthalmic complications, and orbital inflammation. The mechanisms underlying these conditions are complex and not fully understood, complicating diagnosis and treatment. Current management strategies largely rely on corticosteroids and immunomodulatory therapies, with decisions about continuing ICI therapy being highly individualized. Despite progress, there remain substantial gaps in understanding the precise mechanisms, risk factors, and optimal management of OirAEs. Future research should focus on developing predictive biomarkers, refining treatment guidelines, and improving our understanding of how to balance cancer treatment efficacy with the mitigation of OirAEs. Enhanced awareness in clinicians is essential to prevent irreversible ocular damage and to ensure the best possible outcomes for patients undergoing ICI therapy.

## Figures and Tables

**Figure 1 biomedicines-12-02547-f001:**
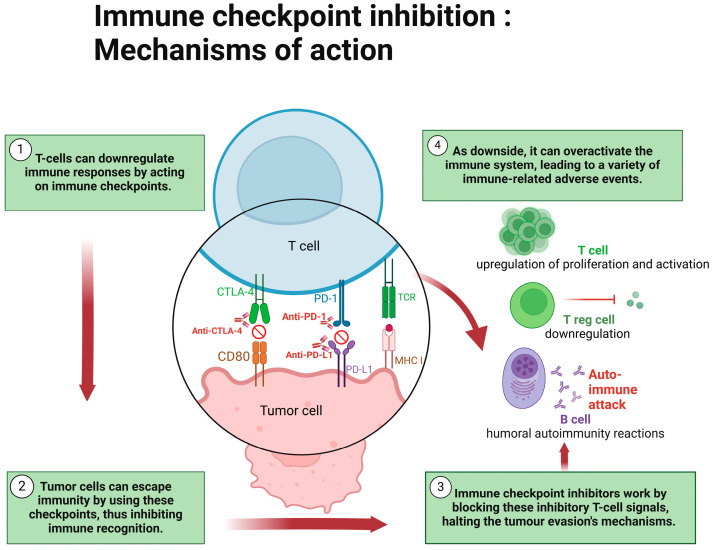
Mechanism of action of immune checkpoint inhibitors. Adapted from Martins et al. and Postow et al. [11,12]. Created with BioRender.com.

**Figure 2 biomedicines-12-02547-f002:**
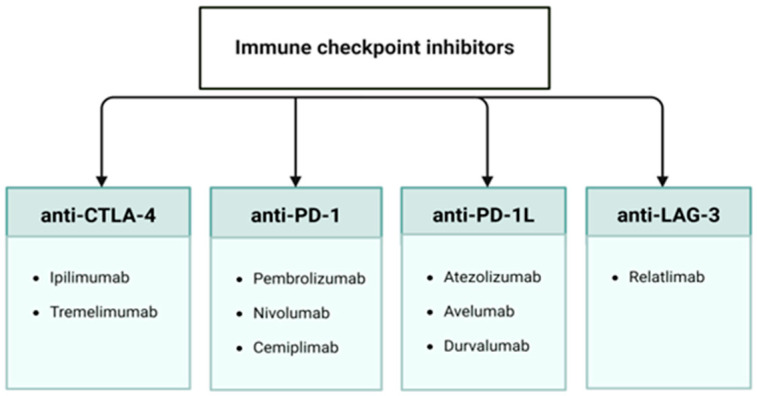
Classes of immune checkpoint inhibitors.

**Figure 3 biomedicines-12-02547-f003:**
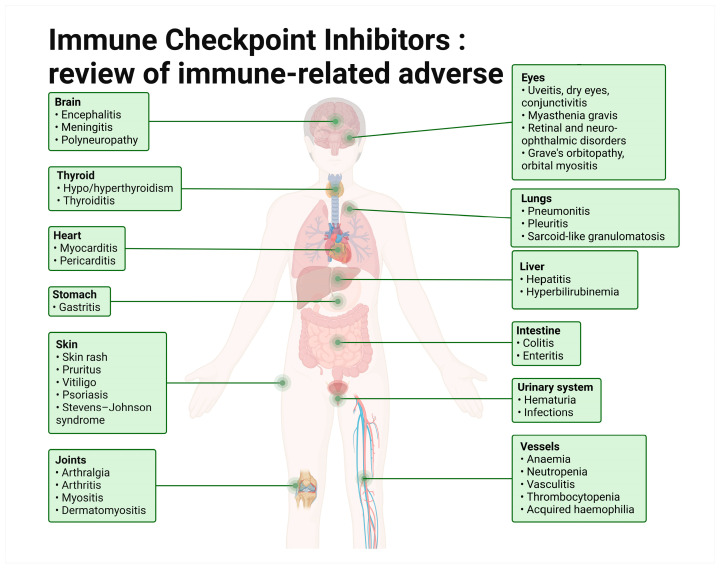
General adverse events associated with ICI use. Adapted from Jamerson Silva’s original template(template available on Biorender.com).

**Figure 4 biomedicines-12-02547-f004:**
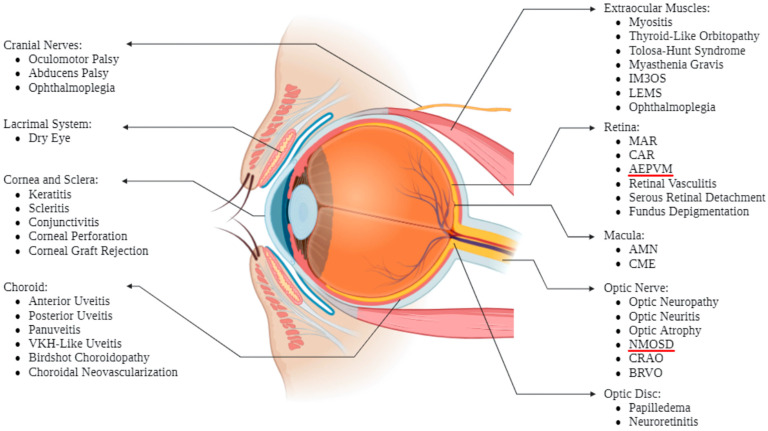
Anatomical localization of OirAEs. Described ophthalmic irAEs are indicated based on anatomical location of the affected structure. Created with BioRender.com. AEPVM = acute exudative paraneoplastic vitelliform maculopathy; AMN = acute macular neuroretinopathy; BRVO = branch retinal vein occlusion; CAR = cancer-associated retinopathy; CME = cystoid macular edema; CRAO = central retinal artery occlusion; IM3OS = Myositis, Myocarditis, and Myasthenia Gravis Overlap Syndrome; LEMS = Lambert–Eaton Myasthenic Syndrome; MAR = melanoma-associated retinopathy; NMOSD = neuromyelitis optica spectrum disorder; VKH = Vogt-Koyanagi-Harada. Created in BioRender.com.

**Figure 5 biomedicines-12-02547-f005:**
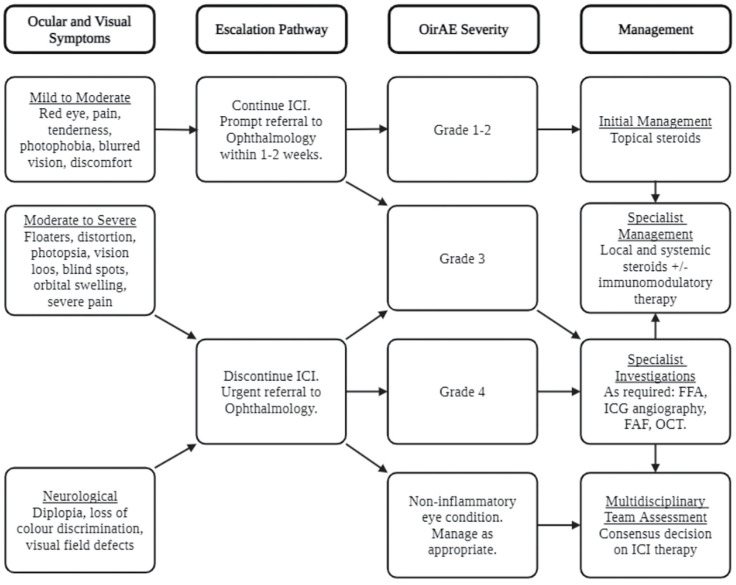
Suggested management algorithm for patients treated with ICIs who develop OirAEs. Adapted from Shahzad et al. [17]. Created with BioRender.com. FAF = fundus autofluorescence; FFA = fundus fluorescein angiography; ICG = indocyanine angiography; OCT = optical coherence tomography. Created in BioRender.com.

**Figure 6 biomedicines-12-02547-f006:**
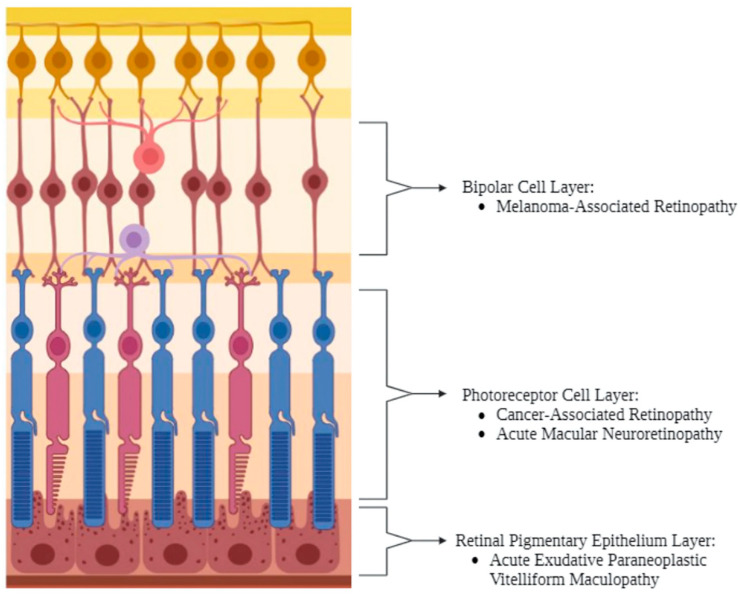
Retinal localization of MAR, CAR, AEPVM, and AMN. ICI-induced retinopathies and maculopathies are indicated based on the targeted layer of the retina. Created with BioRender.com. AEPVM = acute exudative paraneoplastic vitelliform maculopathy; AMN = acute macular neuroretinopathy; CAR = cancer-associated retinopathy; MAR = melanoma-associated retinopathy. Created in BioRender.com.

**Table 1 biomedicines-12-02547-t001:** Select indications of available ICIs. Further information on indication conditions can be found in the cited product monographs.

ICI	Class	Select Indications	Source
Ipilimumab (Yervoy)	Anti-CTLA-4	Unresectable or Metastatic Melanoma; Metastatic Renal Cell Carcinoma; Metastatic Non-Small-Cell Lung Cancer; Microsatellite Instability-High (MSI-H)/Mismatch Repair-Deficient Metastatic Colorectal Cancer; Unresectable Malignant Pleural Mesothelioma; Unresectable or Metastatic Esophageal Squamous Cell Carcinoma	[2]
Tremelimumab (Imjudo)	Anti-CTLA-4	Unresectable Hepatocellular Carcinoma	[5]
Pembrolizumab (Keytruda)	Anti-PD-1	Melanoma; Non-Small-Cell Lung Carcinoma; Hodgkin Lymphoma; Primary Mediastinal B-cell Lymphoma; Urothelial Carcinoma; Renal Cell Carcinoma; Colorectal Cancer; Microsatellite Instability-High Cancer; Endometrial Carcinoma; Head and Neck Cancer; Gastric or Gastroesophageal Junction Adenocarcinoma; Esophageal Cancer; Triple-Negative Breast Cancer; Cervical Cancer; Biliary Tract Carcinoma	[4]
Nivolumab (Opvido)	Anti-PD-1	Unresectable or Metastatic Melanoma; Metastatic Non-Small-Cell Lung Cancer; Unresectable Malignant Pleural Mesothelioma; Metastatic Renal Cell Carcinoma; Squamous Cell Carcinoma of the Head and Neck; Classical Hodgkin Lymphoma; Microsatellite Instability-High/Mismatch Repair-Deficient Metastatic Colorectal Cancer; Gastric Cancer, Gastroesophageal Junction Cancer, or Esophageal Adenocarcinoma; Urothelial Carcinoma; Unresectable or Metastatic Esophageal Squamous Cell Carcinoma	[7]
Cemiplimab (Libtayo)	Anti-PD-1	Cutaneous Squamous Cell Carcinoma; Non-Small-Cell Lung Cancer; Basal Cell Carcinoma; Cervical Cancer	[6]
Atezolizumab (Tecentriq)	Anti-PD-L1	Extensive-Stage Small-Cell Lung Cancer; Non-Small-Cell Lung Cancer; Unresectable or Metastatic Hepatocellular Carcinoma; Locally Advanced or Metastatic Triple-Negative Breast Cancer	[3]
Avelumab (Bavencio)	Anti-PD-L1	Locally Advanced or Metastatic Urothelial Carcinoma; Metastatic Merkel-Cell Carcinoma	[8]
Durvalumab (Imfinzi)	Anti-PD-L1	Urothelial Carcinoma; Locally Advanced, Unresectable, Stage III Non-Small-Cell Lung Cancer; Extensive-Stage Small-Cell Lung Cancer; Locally Advanced or Metastatic Biliary Tract Cancer; Unresectable Hepatocellular Carcinoma	[9]

**Table 2 biomedicines-12-02547-t002:** Criteria for the grading of ocular irAEs. Adapted from Mazharuddin et al. [26].

Grade 1, Mild or Asymptomatic	Grade 2, Moderate	Grade 3, Severe	Grade 4, Life- (or Sight-) Threatening	Grade 5, Death
Asymptomatic or mild symptoms	Symptomatic with mild decrease in VA BCVA > 20/40 or <3 lines of decreased vision from baseline	Symptomatic with marked decrease in VA BCVA < 20/40 or >3 lines of decreased vision from baseline (up to 20/200)	Life-threatening consequences BCVA < 20/200	Death

BCVA = best-corrected visual acuity; VA = visual acuity.

**Table 3 biomedicines-12-02547-t003:** Ocular irAEs of the posterior segment.

Adverse Event	ICI Regimens Reported in Literature	Cancer Types Reported in Literature	Presentation	Treatments Reported in Literature	Outcome	Studies
Unspecified Retinopathy	Atezolizumab Durvalumab Nivolumab Pembrolizumab	NSCLC LUAD Urothelial carcinoma	Symptoms: Vision loss, central and/or peripheral scotoma, photopsia, nyctalopia, floaters ERG: Progressive photoreceptor damage	ICI withdrawal Systemic and topical corticosteroids Reports of cyclosporin, infliximab, rituximab, alemtuzumab, IVIG, PLEX	Variable prognosis, stability of symptoms or improvement in visual function	[21,24,26,82]
Melanoma- Associated Retinopathy	Ipilimumab + Nivolumab Ipilimumab + Nivolumab + Pembrolizumab Ipilimumab Nivolumab Pembrolizumab	Melanoma Uveal Melanoma	Symptoms: Painless visual loss, photopsia, nyctalopia, shimmering, VF defects Antibody profile: anti-GAPDH, anti-arrestin, anti-enolase, anti-TRPM1, anti-carbonic anhydrase II ERG: Impaired bipolar cell function	ICI withdrawal Oral, topical, systemic, and intraocular corticosteroids IVIG Intraocular anti-VEGF	Poor prognosis and irreversible vision loss due to bipolar cell damage and macular scarring	[17,22,36,83,84,85,86,87]
Cancer- Associated Retinopathy	Nivolumab Atezolizumab Pembrolizumab Ipilimumab + Nivolumab Durvalumab	NSCLC SCLC Lung carcinoma Hepatocellular carcinoma Cervical carcinoma Melanoma Endometrial carcinoma	Symptoms: Painless visual loss, photopsia, nyctalopia, shimmering, VF defects Antibody profile: anti-GAPDH, anti-arrestin, anti-enolase, anti-TRPM1, anti-carbonic anhydrase II, anti-aldolase, anti-RAB6, anti-PKM2, anti-recoverin, anti-TULP1 ERG: Impaired rod and cone response	ICI withdrawal Systemic corticosteroids Rituximab Subtenon triamcinolone injections	Poor prognosis due to irreversible vision loss and photoreceptor scarring. Some reports of mild improvement in symptoms	[36,85]
Acute Exudative Paraneoplastic Vitelliform Maculopathy	Ipilimumab + Nivolumab Ipilimumab Nivolumab Pembrolizumab	Melanoma Carcinoma	Symptoms: blurry vision, metamorphopsia, nyctalopia, photopsia Antibody profile: anti-bestrophin-1, anti-arrestin, anti-enolase, anti-transducin-α, anti-peroxiredoxin OCT: Subretinal accumulation of yellow hyperautofluorescent deposits	Corticosteroids +/− ICI withdrawal vs. watchful waiting Subtenon triamcinolone injections	Good prognosis with regression of lesions	[36,84,85,88]
Acute Macular Neuroretinopathy	Atezolizumab	NSCLC Breast cancer	Symptoms: Central or pericentral scotoma with sudden decline in central VA. Triggers: Tends to follow flu-like symptoms or administration of OCP	ICI withdrawal	Poor prognosis, incomplete visual recovery	[36,89]
Retinal Vasculitis	Ipilimumab + nivolumab Ipilimumab Nivolumab Pembrolizumab Durvalumab	Melanoma NSCLC SCLC Myeloid leukemia	Symptoms: Blurry vision, floaters, scotoma, metamorphopsia	ICI withdrawal Oral, topical, systemic corticosteroids Triamcinolone injections	Good prognosis with resolution of symptoms	[21,90,91,92,93]
Choroidal Neovascularization	Ipilimumab + Nivolumab	Melanoma	Symptoms: photopsias, VF loss Antibody profile: variable anti-bipolar and anti-photoreceptor cell antibodies on immunohistochemistry OCT: atrophic chorioretinal lesions	Bevacizumab injection	Stable lesions or resolution	[36,83,84,94]
Serous Retinal Detachment	Nivolumab Pembrolizumab Ipilimumab + Nivolumab	Melanoma Renal cell carcinoma	Symptoms: vision loss, reduced VA, floaters, flashes Tends to occur in the presence of VKH-like uveitis and panuveitis	ICI withdrawal Oral and topical corticosteroids Subtenon triamcinolone injections	Variable, ranging from complete resolution to chronic retinal detachments	[22,26,83,84,95,96,97]
Macular Edema	Ipilimumab + nivolumab Ipilimumab Nivolumab Pembrolizumab	Melanoma	Symptoms: Floaters, blurry vision. Tends to occur in the presence of uveitis	ICI withdrawal Topical and local corticosteroids	Variable, tendency for lesion and symptom resolution	[76,90]
Fundus Depigmentation	Ipilimumab + Nivolumab Nivolumab Pembrolizumab	Ipilimumab + Nivolumab Nivolumab Pembrolizumab	Symptoms: Asymptomatic, but frequent co-presentation with vitiligo and poliosis of eyebrows and eyelashes. Presumed to result from immune-mediated attack against self-melanocytes without the associated inflammation present in VKH-like uveitis	No targeted therapies	Stable lesions or resolution	[36,98,99,100]

ERG = electroretinography; GAPDH = glyceraldehyde 3-phosphate dehydrogenase; IVIG = intravenous immunoglobulins; LUAD = lung adenocarcinoma; NSCLC = non-small-cell lung cancer; OCP = oral contraceptive pill; OCT = optical coherence tomography; PKM2 = pyruvate kinase M2; PLEX = plasmapheresis; RAB6 = Ras-related protein RAB-6A; SCLC = small-cell lung cancer; TRPM1 = transient receptor potential cation channel subfamily M member 1; TULP1 = TUB-like protein 1; VA = visual acuity; VEGF = vascular endothelial growth factor; VF = visual field; VKH = Vogt-Koyanagi-Harada.

**Table 4 biomedicines-12-02547-t004:** Neuro-ophthalmic irAEs.

Adverse Event	ICI Regimens Reported in Literature	Cancer Types Reported in Literature	Presentation	Treatments Reported in Literature	Outcome	Studies
Optic Neuropathy	Ipilimumab + nivolumab Pembrolizumab Ipilimumab Cemiplimab Atezolizumab Durvalumab	Melanoma Prostate cancer HNSCC	Symptoms: painless visual loss, floaters, VF deficits, optic nerve edema Antibody profile: variable, seronegative or anti-recoverin, anti-aldolase, anti-enolase, anti-transducin-α, anti-TRPM1, anti-ANA1 MRI: Variable, no optic nerve defect vs. perineural hyperenhancement of distal segment. Rare optic nerve atrophy	ICI withdrawal Oral, systemic corticosteroids IVIG, rituximab, PLEX, cyclosporine	Good prognosis with improvement of symptoms	[21,53,76,82,94,95,97,98,101,108]
Optic Neuritis	Nivolumab Ipilimumab Ipilimumab + nivolumab Pembrolizumab Atezolizumab	Melanoma NSCLC SCLC LUAD RCC Glioblastoma multiforme	Symptoms: Most commonly bilateral painless reduction in VA with intact color vision. Photopsia, floaters, flashes, smudge, and halo also reported. Antibody profile: seronegative. HVF: Variable defect patterns including altitudinal, arcuate, diffuse, central, and cases of no defect. MRI: Variable, no optic nerve defect vs. perineural hyperenhancement of distal segment	Oral, systemic corticosteroids +/− ICI withdrawal IVIG, rituximab, PLEX, cyclosporine, MMF	Overall good prognosis with improvement, rare cases of worsening symptoms	[21,24,36,53,70,96,97,99,102,109,110]
Neuromyelitis Optica Spectrum Disorder	Nivolumab	Lung squamous cell carcinoma	Symptoms: Blurry vision. Other symptoms not well-described in literature. Antibody profile: anti-aquaporin-4	Corticosteroids PLEX	Stable, minimal response to treatment	[99,102]
Neuroretinitis	Ipilimumab	Melanoma	Symptoms: metamorphopsia, scotoma, pain, redness, photophobia	ICI withdrawal Corticosteroids Brimonidine Timolol	Good prognosis with resolution of symptoms	[109]
Central Retinal Artery Occlusion	*Insufficient reporting from literature*	*Insufficient reporting from literature*	*Insufficient reporting from literature*	*Insufficient reporting from literature*	*Insufficient reporting from literature*	[111]
Branch Retinal Vein Occlusion	*Insufficient reporting from literature*	*Insufficient reporting from literature*	*Insufficient reporting from literature*	*Insufficient reporting from literature*	*Insufficient reporting from literature*	[111]
Myasthenia Gravis	Nivolumab Ipilimumab + nivolumab Pembrolizumab	Melanoma NSCLC Bladder carcinoma Urothelial carcinoma Thymoma RCC	Symptoms: Ptosis, diplopia, ophthalmoplegia. Generalized muscle weakness. Antibody profile: seronegative or anti-AchR	ICI withdrawal Corticosteroids Pyridostigmine IVIG or PLEX as first-line therapy for bulbar symptoms Immune modulation (infliximab, rituximab, MMF, NSAIDs)	Variable, from full resolution to high risk of mortality	[26,36,107,112,113]
Myositis, Myocarditis, and Myasthenia Gravis Overlap Syndrome	Pembrolizumab Nivolumab Cemiplimab Ipilimumab + above Durvalumab + tremelimumab Camrelizumab Sintilimab Durvalumab Atezolizumab	Melanoma Lung SCC Thymoma RCC Bladder cancer Chondroma Mesothelioma Gastric adenocarcinoma	Symptoms: See Myasthenia Gravis. Also associated with palpitations, presyncope, chest pain, and myalgias. Rapidly progressive. Antibody profile: Variable. Less commonly seronegative, more often one or more of anti-AChR, anti-MuSK, anti-titan, anti-ryanodine	Avoid corticosteroids, prioritize immunomodulation (infliximab, rituximab, PLEX). Supportive treatment	Variable High mortality rate	[114,115]
Lambert–Eaton Myasthenic Syndrome	Nivolumab	Lung squamous cell carcinoma	Symptoms: Ptosis, proximal muscle weakness, autonomic dysfunction. Antibody profile: anti-VGCC	ICI withdrawal Corticosteroids Amifampridine Pyridostigmine Carboplatin Etoposide	Good prognosis	[36,109,116,117]
Unspecified Ophthalmoplegia	Pembrolizumab Nivolumab Ipilimumab + nivolumab Durvalumab Sintilimab	RCC NSCLC SCLC Lung squamous cell carcinoma LUAD Leiomyosarcoma	Symptoms: diplopia, ptosis, strabismus. Investigations tend to be normal (MRI, repetitive nerve stimulation, CSF analysis)	ICI withdrawal Oral and IV corticosteroids Pyridostigmine MMF	Good prognosis, with improvement and resolution of symptoms	[24,107,109,118,119]
Cranial Nerve Palsies	Durvalumab Nivolumab	Kidney cancer	Symptoms: Diplopia, ptosis. MRI: Variable. Normal vs. leptomeningeal enhancement	ICI withdrawal Systemic corticosteroids	Good prognosis with improvement and resolution	[36,70,76,107,109,120,121,122]
Opsoclonus	Ipilimumab + nivolumab	Melanoma SCLC Breast cancer Mesothelioma	Symptoms: Involuntary conjugate horizontal eye movements. Frequently presents as part of Opsoclonus–Myoclonus–Ataxia syndrome with encephalitis, dystonia, tremor, and chorea Antibody profile: Variable. Seronegative vs. anti-Ri, anti-SOX-1	Systemic corticosteroids IVIG	Variable prognosis	[109,123]
Temporal Arteritis	Ipilimumab Pembrolizumab Nivolumab Ipilimumab + nivolumab Ipilimumab + bevacizumab	Melanoma NSCLC Pleural mesothelioma	Symptoms: Blurry vision, diplopia, transient vision loss, headache, scalp tenderness, jaw claudication, shoulder myalgia	ICI withdrawal Corticosteroids Tocilizumab	Good prognosis	[24,109]

AChR = acetylcholine receptor; ANNA1 = antineuronal nuclear antibody type 1; IVIG = intravenous immunoglobulins; LUAD = lung adenocarcinoma; MMF = mycophenolate mofetil; NSCLC = non-small-cell lung cancer; PLEX = plasmapheresis; RCC = renal cell carcinoma; SCLC = small-cell lung cancer; SOX-1 = SRY-box transcription factor 1; TRPM1 = transient receptor potential cation channel subfamily M member 1; VA = visual acuity; VGCC = voltage-gated calcium channel; VF = visual field.

**Table 5 biomedicines-12-02547-t005:** Ocular irAEs of the orbit and adnexa.

Adverse Event	ICI Regimens Reported in Literature	Cancer Types Reported in Literature	Presentation	Treatments Reported in Literature	Outcome	Studies
Thyroid orbitopathy	Ipilimumab Nivolumab Pembrolizumab Tremelimumab Ipilimumab + nivolumab Tremelimumab + durvalumab	Melanoma RCC Hepatocellular carcinoma Urothelial carcinoma NSCLC Merkel cell carcinoma Lung adenocarcinoma	Symptoms: Bilateral orbital pain, exophthalmos, periocular swelling and eyelid erythema, conjunctival injection, and diplopia, with restriction in eye movement MRI: variable, most describe diffuse enlargement of all extraocular muscles, most in inferior and medial rectus muscles in a symmetric pattern, and muscle tendon sparring Laboratory findings: variable, with normal to elevated values (TSH, T3, FT4, TSI, anti-TPO) CAS: variable from 1-6	Observation without systemic treatment High-dose intravenous methylprednisolone followed by oral prednisone (tapering) ICI withdrawal in some	Resolution and improvement in most cases A case of persistent bilateral orbitopathy with primary gaze diplopia and ophthalmoplegia is reported	[132,133,134,135,136,137]
Inflammatory orbitopathy	Ipilimumab Pembrolizumab	Melanoma	Symptoms: Acute eye pain, proptosis, diffuse severe ophthalmoparesis, burning, injection, tearing, foreign body sensation, photophobia, and intermittent diplopia MRI: shows enlargement of extraocular muscles, with involvement of tendinous insertions Laboratory findings: negative for thyroid-stimulating immunoglobulins, thyroperoxidase antibodies, and thyroglobulin antibodies	Corticosteroids ICI withdrawal in some	Resolution and improvement	[49,121,122,123]
Orbital myositis/myopathy	Ipilimumab Pembrolizumab Nivolumab + ipilimumab	Melanoma NSCLC Prostate adenocarcinoma Bladder urothelial carcinoma Gastric adenocarcinoma Non-Hodgkin’s lymphoma Gastroesophageal adenocarcinoma Renal, pelvis, and ureter cancer	Symptoms: Bilateral orbital inflammation, painful diplopia and proptosis MRI: extraocular muscles presentation varies, with both normal and abnormal size reported. Imaging studies show tendon-involving myositis in some cases. Laboratory workup: negative for thyroid orbitopathy and other orbital inflammatory processes	IV corticosteroid and IV immunoglobulin therapy Combination of methylprednisolone and mycophenolate mofetil, and, in 1 case, additional medication with IV immunoglobulin High-dose oral steroids ICI withdrawal in some	Variable, with resolution and improvement in most cases, and cases with no improvement and worsening reported	[16,134,138,139,140,141,142,143,144,145]
Orbital apex syndrome	Pembrolizumab Ipilimumab	Pulmonary adenocarcinoma Melanoma	Symptoms: Ptosis, proptosis, vision loss, right RAPD, restricted ocular movement, headache, severe vision loss, diplopia, ophthalmoplegia MRI and laboratory investigations suggest an inflammatory process	Steroid pulse therapy High-dose IV methylprednisolone, then taper oral prednisone ICI withdrawal	Good prognosis with improvement of symptoms. Persistent esotropia	[146,147]
Tolosa–Hunt Syndrome	Ipilimumab	Melanoma	Symptoms: unilateral headache, acute onset of ocular pain, epiphora, double vision, mydriasis, ptosis, paresis of the oculomotor nerve MRI: prominent neural sheath of the optic nerve. Laboratory findings: normal values (TSH, negative thyroid autoantibodies)	High dose steroids (IV methylprednisolone), oral dexamethasone, local radiotherapy ICI withdrawal	Improvement of pain and paresis, but persistence of visual disturbances	[148]

**Table 6 biomedicines-12-02547-t006:** Remaining knowledge gaps related to OirAEs.

How does ICI class influence OirAE development and what are their precise mechanisms?
What risk factors determine who is most likely to develop OirAEs?
Are there specific biomarkers predictive of OirAEs and their severity?
Is there a relationship between ethnicity and OirAEs?
Does sex modulate the immune response observed in OirAEs and, and if so, how?
How can treatment strategies for OirAEs be optimized without compromising underlying tumor treatment?

## Data Availability

Not applicable.

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
