# Peer review of "Emerging Ocular Side Effects of Immune Checkpoint Inhibitors: A Comprehensive Review"

_biomedicines, 2024, doi:10.3390/biomedicines12112547_

Round 1

Reviewer 1 Report

Comments and Suggestions for Authors

The manuscript entitled "Emerging Ocular Side Effects of Immune Checkpoint Inhibitors: A Comprehensive Review" is a good review with sufficient literature support and detailed analysis. I recommend the manuscript for publication after some modifications as suggested below;

1. The abstract looks good periferally, however, if the authors can revise the section by including the examples of OirAEs and corresponding side effects, it would be more accurate.

2. The Keywords need not be in Bold

3. Remove statement like "we identified" as it looks non- scientific 

4. I recommend to include a table containing the details of some immune Checkpoint Inhibitors and thei uses

5. There are few typographic issues in the manuscript, which needs to be corrected during proof reading.

Author Response

Thank you very much for your positive feedback on our manuscript, “Emerging Ocular Side Effects of Immune Checkpoint Inhibitors: A Comprehensive Review.” We are grateful for your insightful comments, which have greatly contributed to enhancing the quality of our work. Below, we address each of your specific suggestions in the revised manuscript:

  1. Abstract Revision: We appreciate your suggestion to enrich the abstract with examples of ocular immune-related adverse events (OirAEs). In response, we have revised this section to include specific examples, which we believe adds precision to the abstract.
  2. Keywords Formatting: Following your recommendation, we have removed the bold formatting from the keywords to ensure uniformity and adherence to scientific standards.
  3. Scientific Wording: We have amended the phrasing to remove non-scientific expressions such as “we identified,” ensuring a more objective tone throughout the manuscript.
  4. Inclusion of a New Table: We have incorporated a new table (now labeled as Table 1) to detail immune checkpoint inhibitors and their clinical applications. This table has been added to page 3 following lines 76/78, as suggested. Consequently, we have adjusted the numbering of the subsequent tables.
  5. Proofreading for Typographic Errors: The manuscript has undergone additional proofreading to address any remaining typographic errors, and we trust it is now improved in clarity and readability.

We sincerely thank you for your valuable input and for the opportunity to revise our manuscript. We hope that our modifications meet your expectations and that the revised manuscript is now suitable for publication.

Kind regards,

Reviewer 2 Report

Comments and Suggestions for Authors

The manuscript presented for reviewing is quite interesting and is focused on various aspects of rare ocular immune-related adverse events (irAEs) developed after antitumor therapy on the basis of immune checkpoints inhibitors (ICIs).

It should be noted that this review is not a single one which has been recently published on this subject. The article by Martens A, et al, 2023 (reference 38) is the nearest to this manuscript according to its content but nevertheless the presented review is worth of close attention of a wide cohort of medical readers.

The review covers the data of the latest results obtained in various countries. More than 70% of the cited works are dated to the last 5 years, and the majority of others are not older than 10 years. The total number of references is rather large: 166 articles. The citing is correct and there are no reasons to suspect plagiarism, though the novelty of the subject does not seem to be too high. The review is written in detail, its volume 2-3 times exceeding those of the previously published similar articles.

The clear structure of the manuscript is its advantage. In the Introduction part the authors prove the importance of the highlighted subject and then sequentially and in detail present the mechanisms of the ICIs action, their usage in cancer patients, and adverse events developed in various organs.  At last they come to ocular complications related to anatomic ophthalmic structures.

The authors discuss all known potential mechanisms of damage developed in ophthalmic structures and honestly mention that there are still many gaps in our knowledge in this area. So, further investigations are necessary for adequate proof of the supposed mechanisms.

The manuscript is well illustrated with distinct Figures and informative Tables which help clear understanding the discussed problems: damage mechanisms, differences in adverse events, impact of medicines with various mechanisms of action on developing complications, treatment methods, and possibility of recovering after toxic reactions.

The obvious value of the review which demonstrates the authors’ deep analysis of the highlighted problems are the questions presented in Table 5 which show the gaps in modern knowledge and, respectively, the directions for future investigations.

The manuscript is written in good scientific English, and presents interest to researchers of various profiles: oncologists, immunologists, ophtalmologists, not only to physicians but to experimental scientists as well.

Minimum correction is required for Reference #52 due to absence of the year of publication in it.

Author Response

Thank you for your thorough and thoughtful review of our manuscript. We are grateful for your positive remarks and the constructive feedback you provided. We have addressed each of your specific comments in our updated manuscript and would like to highlight the revisions made to ensure clarity and completeness.

  1. Reference Correction: We appreciate your attention to detail in pointing out the missing publication year in Reference #52. The year of publication has been added, and the reference has been corrected as suggested.
  2. Tables and Figures: We value your positive feedback on our Tables and Figures, and we made minor formatting adjustments to ensure consistency and clarity.
  3. Clarity and Readability: Based on your comments regarding the manuscript’s length, we have streamlined certain sections for brevity while retaining all critical information. We believe this revision enhances the readability and flow of the manuscript for a broader audience.

We appreciate your encouragement and careful review, which greatly contributed to improving our manuscript. Thank you for recognizing the importance of this topic and for your support in making our work accessible and valuable to researchers and clinicians alike.

Sincerely,

Reviewer 3 Report

Comments and Suggestions for Authors

In princuple, I find this review rather straightforward and logically built up. There is a clear storyline and the eye-related complications covered are kind of sorted by severity, and by potential mechanism. There is nothing that requires severe shortening or omission. There is also ample documentation and citations/references added to the review article; Im surprised there is such a alrge body of literature already available in the public domain that deals with these relatively rare complications. But they appear well researched and add to the quelity of the manuscript. 

Overall, the article is already quite long (34 pages in total), but partly because there are lengthy, voluminous tables included. These are a bit difficult to read and handle, and Im not sure if they really do the article much of a favour. 

I dont know if this is a realistic comment, but wouldnt it be helpful to have some images from selected eye-related diseases/complications, that illustrate how these adverse events actually look ? That may be a very helpful addition for identifying patients at risk, and if the article is published and ends up in the public domain, it may even be helpful for patients themselves, helping them to identify potential side effects related to their therapy. 

Figure 1 is nice, but maybe a bit too much text (which could also be in the figure legends). 

Figure 2: maybe the writing is a bit small, and it could be considered to increase font sizes. 

Figures 3 to 5:  the same probably applies to figure 3, 4 and 5 as well, which all should be made larger - maybe full page size even, considering the small font size for the text and the total number of text elements in the figures. (Figure 5 is actually a table and maybe its better to use it as a table, not a figure). The authors may also have to check for resolution especially of the text elements in the uploaded figure files. These are not vector graphics = text will look blurry. That particularly applies to Fig. 5 as well which is nothing but text. BTW: do you really need to include the "created by biorender" full logo in the figures? Isnt it enough to say in the legend? And again, concerning Fig. 5, whats the use of biorender in creating text boxes? 

Some of the sentences (for example line 159 - 165) are rather long and convoluted, which usually doesn't help with comprehension of the text. Just split in 2-3 shorter sentences. 

Tables 1-4 tend to be a bit large and voluminous and may not be so easy to read, when published; but you will see. 

I really like the short and concise Table 5 that summarized unresolved issues in the field! Very nice! 

Overall, I think the manucript reads really well and logical and it immuminated me (as reviewer) on a completely unknown topic (to me) - illustrating that the article is definitely informative and also reads well. Mostly, simple sentences and clear text structure. 

I think it requires at best minor improvements. 

Comments on the Quality of English Language

there are only very few convoluted sentences in the article, otherwise, the English language is really good and offers little for objection or improvement. 

Author Response

Thank you for your thoughtful and encouraging comments regarding our manuscript. We truly appreciate your recognition of the logical structure and comprehensive nature of the content. Below, we have provided specific responses to each of your insightful suggestions.

  1. Inclusion of Images for Eye-Related Diseases/Complications: We appreciate your valuable suggestion on adding images to illustrate the discussed complications. Unfortunately, due to the lack of permissions from published sources with suitable images and the absence of relevant images in our personal database, we are unable to include them at this time.
  2. Figure 1 Text Adjustment: We have revised Figure 1 to be more concise by reducing the text and incorporating additional details into the figure legend, enhancing its clarity and presentation.
  3. Figure 2 Font Size: The font size in Figure 2 has been increased to improve readability and overall aesthetic appeal.
  4. Figures 3 to 5 Formatting and Resolution:
    • All figures have been resized and adjusted for better readability and visual impact.
    • We have converted Figure 5 to a table, now Table 5, following your recommendation.
    • The "created by Biorender" logo is included in compliance with Biorender’s guidelines, as we understand it to be a required element.
    • Regarding resolution, we utilized the highest resolution permissible within Biorender’s platform.
  5. Sentence Structure Improvement (Lines 159-165): This section has been restructured into shorter, more direct sentences to enhance clarity and readability.
  6. Tables 1-4 Readability: We acknowledge your feedback on the table length and complexity. The tables are intentionally detailed to convey essential information as concisely as possible. We will consult with the editor on potential adjustments to formatting upon publication to further optimize readability.

Thank you once again for your constructive feedback and for helping us refine the quality of our manuscript. We are hopeful that these revisions will meet with your approval and contribute to an even stronger final version.

Warm regards,

Reviewer 4 Report

Comments and Suggestions for Authors

The review is very well written, comprehensive, and presents the topic in an insightful manner. The figures and tables are well-designed and only require minor improvements (see minor points below). Apart from these small issues, I congratulate the authors on their excellent manuscript.

Minor points:

  • Figure 1: All lines connecting "immune checkpoint inhibitors" to the four boxes listed below should terminate at the center of each box, as is already the case with Box 4 ("anti-LAG-3").
  • Table 1: All abbreviations used should be explained in a legend accompanying the figure.
  • Line 272: It would be helpful to specify the exact dosage of cyclosporine eye drops and to clarify which type of artificial tear should be chosen (e.g., more lipid-based or more aqueous-based formulations, etc.).

Author Response

Thank you very much for your kind and positive feedback on our manuscript. We are grateful for your appreciation and constructive suggestions, which have greatly contributed to enhancing the clarity and completeness of our work. Please find our responses below to each of your helpful comments.

  1. Figure 1 Connections: We have adjusted the connections in Figure 1 to ensure all lines terminate at the center of each respective box, including Box 4 ("anti-LAG-3"), to improve visual consistency.
  2. Table 1 Abbreviations: A legend has been added below Table 1, providing explanations for all abbreviations used, as per your recommendation.
  3. Line 272 Dosage and Artificial Tear Specification: We have specified the dosage for cyclosporine eye drops and clarified the type of artificial tears (lipid-based formulation) to guide treatment choice more precisely.

Thank you once again for your encouraging comments and invaluable feedback. We hope these revisions meet with your approval and contribute to a stronger final manuscript.

Warm regards,